# Fibronectin Functions as a Selective Agonist for Distinct Toll-like Receptors in Triple-Negative Breast Cancer

**DOI:** 10.3390/cells11132074

**Published:** 2022-06-30

**Authors:** Anthony Ambesi, Pranav Maddali, Paula J. McKeown-Longo

**Affiliations:** Department of Regenerative & Cancer Cell Biology, Albany Medical Center, 47 New Scotland Avenue, Albany, NY 12208-3479, USA; ambesia@amc.edu (A.A.); maddalp@amc.edu (P.M.)

**Keywords:** fibronectin, breast cancer, toll-like receptor

## Abstract

The microenvironment of tumors is characterized by structural changes in the fibronectin matrix, which include increased deposition of the EDA isoform of fibronectin and the unfolding of the fibronectin Type III domains. The impact of these structural changes on tumor progression is not well understood. The fibronectin EDA (FnEDA) domain and the partially unfolded first Type III domain of fibronectin (FnIII-1c) have been identified as endogenous damage-associated molecular pattern molecules (DAMPs), which induce innate immune responses by serving as agonists for Toll-Like Receptors (TLRs). Using two triple-negative breast cancer (TNBC) cell lines MDA-MB-468 and MDA-MB-231, we show that FnEDA and FnIII-1c induce the pro-tumorigenic cytokine, IL-8, by serving as agonists for TLR5 and TLR2, the canonical receptors for bacterial flagellin and lipoprotein, respectively. We also find that FnIII-1c is not recognized by MDA-MB-468 cells but is recognized by MDA-MB-231 cells, suggesting a cell type rather than ligand specific utilization of TLRs. As IL-8 plays a major role in the progression of TNBC, these studies suggest that tumor-induced structural changes in the fibronectin matrix promote an inflammatory microenvironment conducive to metastatic progression.

## 1. Introduction

Breast cancer is the most prevalent cancer in women and the second leading cause of cancer death. TNBC accounts for nearly 20% of all breast cancers and there are currently no approved targeted therapies for treatment [1]. Progression of solid tumors, including TNBC, occurs in a microenvironment characterized by fibrosis and inflammation, thus creating the tissue rigidity integral to tumor growth and metastasis [2,3]. In response to paracrine signals from the tumor cells, stromal fibroblasts differentiate into highly contractile myofibroblasts. These cancer-associated fibroblasts (CAFs) assemble an extracellular matrix (ECM) enriched for the alternatively spliced isoform of fibronectin which contains an extra Type III Domain, Extra Domain A (EDA) [4,5]. Structurally, fibronectin is organized into individually folded domains, termed Types I, II, and III. Unlike the Type I and II domains, the Type III domains are not stabilized by disulfide bonds and unfold in response to increased mechanical force [6,7]. Several studies have shown fibronectin in the breast cancer stroma to be highly stretched due to the unfolding of Type III domains [8,9,10]. How these changes in fibronectin structure impact the tumor microenvironment is not well understood.

EDA fibronectin has been identified as a damage-associated molecular pattern molecule (DAMP) (reviewed in [11]). DAMPs are endogenous molecules, derived from the extracellular matrix or released from damaged cells, which mediate sterile inflammation in a variety of cell types through the activation of Pattern Recognition Receptors (PRRs) [12,13,14,15,16]. The fibronectin-derived DAMPs, FnEDA and FnIII-1c, have been identified as agonists for TLRs, which represent a sub-family of PRRs. TLRs also recognize and are activated by exogenous molecules called pathogen-associated molecular pattern molecules (PAMPs), such as bacterial lipopolysaccharide (LPS), flagellin, lipoproteins, or viral RNA [17,18]. TLRs represent a ten-member family of pattern recognition receptors which function as homo- or heterodimers to regulate an innate immune response following tissue damage or pathogen invasion [19,20]. The EDA domain of fibronectin, either as the individual Type III domain or within the context of the complete protein, causes the homodimerization of TLR4 and the activation of NFκB and MAP kinases to initiate transcription of inflammatory genes in both immune and non-immune cells of the tissue [15,21,22,23,24]. We have also identified a second Type III domain (III-1) of fibronectin which functions as a DAMP. This domain, when partially unfolded (III-1c), initiates an innate immune response in fibroblasts and lung cancer cells by activating either TLR4 or TLR2 [25,26,27,28,29].

In the present study, we evaluated the impact of the two DAMP domains of fibronectin on the release of the pro-tumorigenic cytokine, IL-8, by two TNBC cell lines: the basal-like 1 type, MDA-MB-468, and the mesenchymal stem-like type, MDA-MB-231 [30]. IL-8 plays a significant role in the progression of TNBC by facilitating various steps in tumor progression. In addition to its well-recognized role in driving tumor angiogenesis, IL-8 also promotes tumor metastasis and chemoresistance (reviewed in [31,32]). Recent reports have provided additional support for IL-8′s role in TNBC progression. These studies implicate IL-8 in the development of bone, brain, and lung metastasis [33,34,35], epithelial mesenchymal transition [36], and expansion of cancer stem cells [37]. Our current findings indicate that only the EDA domain of fibronectin activated NFκB signaling and cytokine release in MDA-MB-468 cells, while both the EDA and III-1c domains elicited the response in MDA-MB-231 cells. Unexpectedly, we found that in MDA-MB-468 cells, the release of IL-8 in response to the EDA domain was completely dependent on TLR5, the canonical receptor for flagellin. EDA stimulation of IL-8 release in MDA-MB-231 cells depended on both TLR5 and TLR2, while IL-8 release in response to III-1c was exclusively regulated by TLR2. These data provide the first demonstration of TLR5 activation by an ECM-derived DAMP. Additionally, the data also indicate that activation of specific TLRs by fibronectin DAMPs are not exclusively ligand-specific but also depend on cellular context.

## 2. Materials and Methods

### 2.1. Antibodies and Reagents

The following antibodies were obtained from Cell Signaling (Danvers, MA, USA): phospho-NF-kB p65 (Ser536), phospho-IKKα/β (Ser176/180), phospho-TAK1 (Thr184/187), toll-like receptor 2 (TLR2), and GAPDH. Antibodies for toll-like receptor 5 (TLR5) expression analysis were obtained from Proteintech (Rosemont, IL, USA). TLR2 neutralizing antibodies and the TLR5 inhibitor, TH1020, were from R&D Systems (Minneapolis, MN, USA). TLR5 neutralizing antibody and TLR agonists (flagellin and Pam3CSK4) were obtained from InvivoGen (San Diego, CA, USA). The TLR2 inhibitor, TLR2-IN-C29, was obtained from Abcam Co. Ltd. (Cambridge, MA, USA). Recombinant fibronectin DAMPs, FnEDA, and FnIII-1c, were prepared and purified as previously described [28,38].

### 2.2. Cell Culture

Human dermal fibroblasts (A1-F) and human breast adenocarcinoma cell lines, MDA-MB-468 (ATCC; HBT-132) and MDA-MB-231 (ATCC; CRM-HTB-26), were maintained in a complete medium [Dulbecco’s modified eagle medium (DMEM), Invitrogen/Life Technologies, Corp., Grand Island, NY, USA) supplemented with 1% Pen-Strep (Gibco), 1% GlutaMAX (Gibco) and 10% fetal bovine serum (FBS; Hyclone Laboratories, Logan, UT, USA)] in a humidified chamber at 37 °C/8% CO_2_.

### 2.3. Cell Treatments

Unless otherwise stated, cells were plated onto 48-well culture plates (3 × 10^4^ cells per well) in complete medium, cultured overnight, then rinsed once with 0.1% BSA/DMEM [serum-free medium; DMEM containing 0.1% bovine serum albumin (BSA; Roche Applied Science, Indianapolis, IN, USA), 1% Pen-Strep, 1% GlutaMAX, 1X non-essential amino acids (NEAA; Gibco) and 10 mM HEPES (Gibco)] prior to all treatments. Treatment with inhibitors or blocking antibodies was typically carried out in serum-free medium for 1 h prior to the addition of fibronectin DAMPs (FnEDA and FnIII-1c) and TLR PAMPs (flagellin and Pam3CSK4). Specific treatment times are described in the Figure Legends. Unless otherwise stated, experiments where conditioned medium was collected and used for IL-8 expression analysis, cells were treated for 4 h. IL-8 was measured using a human enzyme-linked immunosorbent assay (ELISA) kit (BD Biosciences, San Diego, CA, USA) according to the manufacturer’s recommended procedure. To capture phosphorylated intermediates of the NFκB signaling pathway, cells were treated with DAMPs and PAMPs for 1 h prior to lysis.

### 2.4. Protein Analysis Using the Wes-ProteinSimple System

Whole-cell lysates were obtained by first rinsing cell layers twice with ice-cold phosphate-buffered saline (PBS) containing 1 mM sodium ortho-vanadate followed by direct lysis in SDS-containing sample buffer (62.5 mM Tris, pH 6.8, 2% SDS, 10% glycerol, 50 mM dithiothreitol, and 0.01% bromophenol blue). Samples were denatured by heating at 95 ˚C for 6 min and an automated capillary-based western system (Wes-ProteinSimple, San Jose, CA, USA) was used for protein detection according to the manufacturer’s recommended procedure. Quantitative analysis of targeted proteins was performed with Compass system software (Wes-ProteinSimple) and quantified relative to GAPDH expression levels.

### 2.5. Suppression of TLR5 and TLR2 Expression

Small interfering RNAs (siRNA) targeting TLR2 (ON-TARGETplus Human TLR2 siRNA SMARTpool) and TLR5 (ON-TARGETplus Human TLR5 siRNA) and non-targeting control siRNA (ON-TARGETplus Non-targeting Control #2) were obtained from Horizon Discovery Ltd. (Lafayette, CO, USA). Typically, cells were plated onto 24-well culture plates (1.5 × 10^4^ cells per well) in complete medium without antibiotics and allowed to attach overnight. Cell layers were then rinsed with Opti-MEM (ThermoFisher Scientific, Waltham, MA, USA) and transfected with siRNAs (40 nM; 400 µL) in Opti-MEM containing DharmaFECT2 transfection reagent (Horizon Discovery Ltd. LaFayette, CO, USA). After 4 h, 400 µL of 20%FBS, DMEM was added. Transfected cells were cultured for 2–4 days prior to treatment as described above. TLR expression analysis using an automated western capillary-based western system (WES-ProteinSimple) was used to confirm TLR knockdown.

### 2.6. Statistical Analysis

Quantitative results were reported as mean ± standard error of the mean (SEM). A Student’s *t*-test was used to compare the levels of the two experimental groups and a One-Way ANOVA was used for comparison over the two groups. The significance level was set at 0.05 and SigmaPlot 12.5 (Systat Software, Chicago, IL, USA) was used for statistical analysis.

## 3. Results

In earlier studies we have shown that the addition of the fibronectin-derived DAMPS, FnEDA and FnIII-1c, to human embryonic skin fibroblasts, A1-F, stimulated the TLR4/NFκB-dependent release of the proinflammatory cytokines, IL-8 and TNFα [25,27,39,40]. In the current study, we evaluated the effect of these fibronectin DAMPs on cytokine release in two subtypes of triple-negative breast cancer cells, MDA-MB-468 (basal-like1) and MDA-MB-231 (mesenchymal stem-like). As we have reported previously [27], the fibronectin-derived DAMPs, FnEDA and FnIII-1c, induced a dose-dependent release of IL-8 in dermal fibroblasts (Figure 1A). When added individually to fibroblasts, each DAMP induced similar amounts of IL-8 (approximately 0.5 ng/mL). However, when added together, the DAMPs induced levels of IL-8 which were synergistic (2.5 ng/mL) rather than additive. A different response was seen in MDA-MB-468 cells which released approximately 20-fold higher levels of IL-8 (50 ng/mL) but only in response to FnEDA (Figure 1B). The MDA-MB-468 cells did not release IL-8 in response to FnIII-1c. A third type of response was seen in the MDA-MB-231 cells. As shown in Figure 1C, the Fn DAMPS individually induced approximately 10-fold higher amounts of IL-8 (4–6 ng/mL) when compared with the A1-F skin fibroblasts. In contrast to the skin fibroblasts, the IL-8 amounts released by the MDA-MB-231 cells in response to both DAMP domains were additive (approximately 10 ng/mL) and not synergistic. These findings indicate that the innate immune response to fibronectin-derived DAMPs is cell type-specific, both with respect to the DAMPs recognized and the amount of cytokine released.

Previous studies in numerous model systems and cell types have consistently demonstrated that the activation of the innate immune response by the FnEDA domain is mediated through the TLR4-dependent activation of NFκB [24,41,42,43,44,45,46,47,48,49]. However, after testing several neutralizing antibodies to various TLRs including TLR4 (data not shown), we found that in MDA-MB-468 cells, IL-8 release in response to FnEDA was entirely dependent on TLR5 (Figure 2A). TLR5 is the receptor which recognizes the bacteria-derived PAMP, flagellin. This neutralizing antibody was equally effective at blocking the IL-8 response to FnEDA as it was to TLR5′s cognate ligand, flagellin (Figure 2B). In MDA-MB-468 cells, the neutralizing antibody to TLR2 had no effect on IL-8 release in response to either FnEDA or flagellin and served as a negative control (Figure 2A,B). In contrast, cytokine release in response to both FnEDA (Figure 2C) and FnIII-1c (Figure 2D) was regulated by TLR2 in MDA-MB-231 cells. Interestingly, blocking antibodies to TLR2 and TLR5 inhibited approximately 70% and 30% respectively of the IL-8 released in response to FnEDA (Figure 2C). When added together, the two antibodies completely prevented IL-8 release in response to FnEDA (Figure 2C). In contrast, induction of IL-8 by FnIII-1c was entirely dependent on TLR2 (Figure 2D).

TLRs activated by DAMPs are known to stimulate innate immune responses through activation of the NFκB pathway [50]. As shown in Figure 3, immuno-analysis of proteins using the Wes system showed that FnEDA treatment of MDA-MB-468 cells resulted in the phosphorylation of NFκB (panel A) as well as its immediate upstream activators, IKKα/β (Figure 3B) and TAK1 (Figure 3C). A similar activation was seen with flagellin, which served as a positive control. All phosphorylation events were inhibited in the presence of blocking antibody to TLR5. Consistent with the results shown in Figure 1B, there was no activation of the NFκB pathway by FnIII-1c. These data were quantified in Figure 3D–F. A similar outcome was seen when MDA-MB-468 cells were pretreated with the chemical inhibitor for TLR5, TH1020 [51]. As shown in Figure 4A, TH1020 inhibited the induction of IL-8 by both FnEDA and flagellin, with similar kinetics. TH1020 also blocked the activation of the NFκB pathway in response to FnEDA and flagellin. Wes-protein analysis of the signaling intermediates downstream of TLR5 indicated a significant decrease in the phosphorylation of NFκB (Figure 4B) and its upstream activators, p-IKK α/β and p-TAK1 (Figure 4C,D). These data are quantified in Figure 4E–G.

To confirm the role of TLR5 in the activation of the innate immune response by Fibronectin-derived DAMPs, TLR5 was knocked down in MDA-MB-468 cells using siRNA. As shown in Figure 5, partial knockdown of TLR5 (panels A and B) resulted in a significant decrease in IL-8 expression in response to both FnEDA and flagellin (panel C). Consistent with the inhibition of IL-8 release, siRNA to TLR5 also inhibited the phosphorylation of NFκB, IKK and TAK1 in response to both FnEDA and flagellin (panels D–F). These data are quantified in panels G-I and provide additional evidence that TLR5 may regulate sterile inflammation during the progression of TNBC. To our knowledge, this is the first time TLR5 has been shown to activate the NFκB pathway and induce cytokine release in response to an ECM-derived DAMP.

In MDA-MB-231 cells, TLR5 also contributed to the innate immune response to DAMPs, although to a lesser extent than that seen in MDA-MB-468 cells. As shown earlier in Figure 2C, FnEDA induced expression of IL-8 was partially (approximately 30%) dependent on TLR5, with the remainder dependent on TLR2. As shown in Figure 6A, Wes protein analysis indicated that phosphorylation of NFκB in response to FnEDA was reduced to nearly control levels when MDA-MB-231 cells were pretreated with neutralizing antibodies to both TLR2 and TLR5. Similar results were seen for the upstream activators of NFκB, IKK and TAK1 (Figure 6B,C). These data, which are quantified in Figure 6D–F, showed that neutralizing antibodies to TLR2 and TLR5 significantly inhibited activation of the NFκB signaling pathway in response to FnEDA. In contrast, release of IL-8 from MDA-MB-231 in response to FnIII-1c was shown to depend entirely on TLR2 (Figure 2D). Figure 6G–I shows activation of the NFκB pathway by FnIII-1c, as demonstrated by increased phosphorylation of the NFκB subunit, p65 and its upstream activators, IKK and TAK1. These phosphorylation events were all significantly inhibited by pretreatment of MDA-MB-231 cells with the TLR2 neutralizing antibody. These data are quantified in Figure 6J–L. To characterize further the role of TLR2 in FnIII-1c mediated cytokine release, TLR2 was knocked down using siRNA.

As shown in Figure 7A,B, Wes analysis indicated a significant decrease in the expression of TLR2 in the MDA-MB-231 cells treated with the siRNA. As indicated in Figure 7C, IL-8 expression in the knockdown cells was significantly decreased in response to both FnEDA and FnIII-1c. Consistent with previously shown data (Figure 2C,D), siRNA to TLR2 completely inhibited the IL-8 response induced by FnIII-1c while partially reducing IL-8 release in response to FnEDA. siRNA to TLR2 also significantly decreased the cellular response to the canonical TLR2 ligand, Pam3CSK4 (Pam3), which served as a positive control (Figure 7C). Similar findings were obtained using the chemical inhibitor of TLR2 [52], C29 (Figure 7D). C29 was completely effective in preventing IL-8 release in response to Pam3 and FnIII-1c and only partially effective in blocking IL-8 release in response to FnEDA. C29 had little effect on IL-8 release in response to flagellin. Taken together, these data indicate that in MDA-MB-231 cells, the activation of the innate immune response to fibronectin DAMPs is mediated through TLR2 and TLR5. IL-8 release in response to FnIII-1c depended entirely on TLR2 while IL-8 release in response to FnEDA depended on both TLR2 and TLR5.

## 4. Discussion

The fibronectin matrix is known to play a significant and varied role in the progression of breast cancer, although the specific molecular mechanisms are not well understood. The signaling networks between the stromal and cancer cells within the tumor are exceedingly complex and interdependent. This interplay takes place alongside a background of poorly understood mechanically activated signals which are triggered in response to the increase in tissue rigidity [53,54,55]. As part of this reprogramming of the stromal microenvironment by the tumor, the ECM is enriched by the EDA isoform of fibronectin [56]. This isoform, also referred to as oncofetal fibronectin, is generated through alternative splicing and not typically seen in the blood plasma or in adult tissues except during wound repair or in association with disease processes [57,58]. The fibronectin matrix is thought to regulate several steps in breast cancer progression, including tumor growth, invasion, survival, dormancy, and resistance to chemotherapy [59]. As the EDA isoform of fibronectin is not present in normal adult tissue, the increase in the EDA isoform within the tumor stroma has been proposed as an indicator of disease progression and as a potential therapeutic target [60].

The mechanically sensitive stromal fibronectin matrix is continually remodeled by the highly contractile CAFs which stretch and unfold fibronectin’s Type III domains [7], thus rearranging the availability of biologically active sites [10]. In the current study, we have used a fibronectin peptide, FnIII-1c, which represents a stable intermediate structure predicted to form in response to increased contractile force [61]. How the availability of this fibronectin matrix DAMP is regulated in the tumor microenvironment is not known but may involve an increase in tissue mechanical force or activation of proteases [62]. A model is shown in Figure 8.

In the current study, we show that the EDA and III-1c domains of fibronectin induce the TLR-dependent release of the pro-tumorigenic cytokine, IL-8, by two TNBC cells lines. Accumulating evidence now points to IL-8 as playing a major role in the progression of breast cancer. Delineating the pathways which contribute to the control of prevailing IL-8 levels at various stages of breast cancer progression may provide novel targets for therapeutic intervention. IL-8 influences breast tumorigenesis by promoting invasivity [65], epithelial–mesenchymal transition (EMT) [66], mesenchymal and stem cell phenotypes [67,68,69], angiogenesis [70], chemoresistance, and suppression of anti-tumor immunity [71]. The presence of IL-8 in breast tissues is negatively correlated with disease outcome and survival [72,73]. Recent findings have now implicated the inflammatory cytokine IL-8 in the induction of breast cancer emergence from dormancy [74], indicating a role for this cytokine in the re-emergence of actively growing cancer.

TLRs are expressed in a variety of cancers including breast cancer where they have been proposed as both biomarkers to monitor breast cancer progression and as targets for therapeutic intervention [75]. In the case of TLR5, studies have shown both pro- and anti-tumorigenic effects. The canonical ligand for TLR5, flagellin, was shown to slow breast cancer cell proliferation rates as well as rates of tumor growth in mouse xenograft models [76]. On the other hand, overexpression of TLR5 in breast tumors has been linked to increased metastasis and TLR5 polymorphisms have been linked to breast cancer susceptibility [77]. In another study, flagellin was found to inhibit breast cancer cell growth by enhancing autophagy [78]. The flagellin derivative, entolimod, has been used as a modulator of innate immunity to inhibit tumor metastasis by stimulating the recruitment of cytotoxic T-lymphocytes [79,80,81]. Although numerous studies have shown that the FnEDA domain activates innate immune responses through TLR4, FnEDA dependent IL-8 release from the MDA-MB-468 cells was entirely dependent on TLR5. FnIII-c did not activate TLR signaling in MDA-MB-468 cells. Previously, flagellin was thought to be the sole agonist for TLR5, but more recent studies have shown that the nuclear protein, HMGB, can also serve as a TLR5 DAMP [82]. In the current study, we now identify FnEDA as the first ECM-derived agonist for TLR5.

In TNBC, TLR2 signaling has been shown to exhibit both pro- and anti-tumorigenic effects by inhibiting tumor growth [83] and promoting lung metastasis [84]. Circulating levels of TLR2 are associated with an increased risk of breast cancer [85]. When added to the MDA-MB-231 cells, FnIII-1c stimulated an innate immune response by activating TLR2. In earlier work, we have shown that the FnIII-1c domain can elicit TLR4- or TLR2-dependent innate immune responses in skin and lung fibroblasts, respectively [25,26]. In contrast to the findings in MDA-MB-468 cells, FnEDA induced IL-8 release by MDA-MB-231 cells was dependent on both TLR2 and TLR5.

Regulation of TLR signaling is complex as it typically involves co-receptors and accessory proteins that can positively or negatively regulate more than one TLR (reviewed in [86,87]). In addition, TLR activity can be modulated by parallel signaling pathways initiated by other ligand–receptor interactions [24,88,89,90]. Although FnEDA is well recognized as an agonist for TLR4, the co-receptors or accessory molecules which might contribute to this activation are not well characterized. The prototypic ligand for TLR4 is bacterial LPS. LPS does not bind directly to TLR4 but interacts with an LPS binding molecule (LBP) and TLR4 accessory proteins, CD14 and MD2, which are required for TLR4 activation and downstream signaling in response to LPS [91]. Interestingly, LBP has also been shown to bind bacterial lipopeptide and peptidoglycan and direct the formation and activation of the TLR2 heterodimers, TLR2/1 and TLR2/6 [92]. Similarly, CD14, has been shown to function as a co-receptor for several TLRs and has been proposed as a therapeutic target to modulate the immune responses [93]. Whether these proteins, or others, contribute to the activation of TLR4, TLR2, or TLR5 by FnEDA is an important topic for future studies.

The α4β1 integrin has been shown to regulate the TLR4-mediated induction of IL-8 by FnEDA in skin fibroblasts [39]. It is not known whether α4β1 exerts its effects on TLR4-mediated IL-8 release by activating a parallel signaling pathway which transactivates the TLR or whether α4β1 forms a pre-existing complex with TLR4 on the cell surface. Several integrins bind EDA including α4β1, α9β1, and α4β7. Further studies are needed to understand what role they may play in regulating the TLR-mediated immune response to FnEDA. Interestingly, trans retinoic acid has been shown to enhance the TLR5 immune response to flagellin through the upregulation of CD14 [94].

It is becoming increasingly clear that the ligand specificity of TLRs can vary among cell types. How the fibronectin DAMPs selectively activate different TLRs is not well understood. Accessory proteins or co-receptors which regulate TLR affinities and subcellular localizations are believed to play a role in ligand discrimination by specific TLRs [95]. In an earlier study using an HEK expression system, we have shown that the activation of TLR4 by FnIII-1c requires the TLR4 accessory molecules, MD2 and CD14 [28]. Whether these same accessory molecules are required for the TLR2 activation by FnIII-1c in MDA-MB-231 cells is not known, but CD14 has been shown to function as an accessory molecule for the activation of TLR2 [96]. We have also shown previously that the FnEDA binding integrin, α4β1, modulates TLR4 signaling in skin fibroblasts [39]. Interestingly, another EDA binding integrin receptor, α9β1, is overexpressed in TNBC and associated with reduced overall survival and tumor invasion [5,97]. Additionally, reagents designed to disrupt integrin binding to the EDA domain of fibronectin inhibit breast cancer progression in mouse models [4]. The findings reported herein, point to the need for future studies to define the molecular basis for the cell type selection of TLR activation by tumor-derived DAMPs. Such studies may help to define subtype-specific targets for novel therapies to treat TNBC.

## Figures and Tables

**Figure 1 cells-11-02074-f001:**
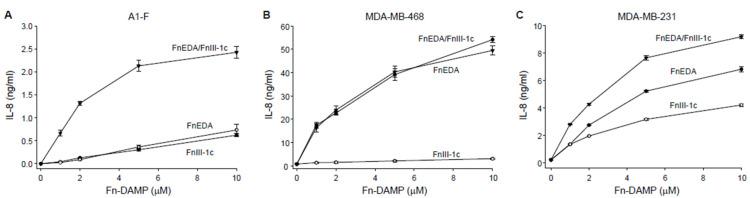
FnEDA and FnIII-1c stimulate IL-8 synthesis in human dermal fibroblast and triple-negative breast cancer cells in a dose-dependent manner. (**A**) Human dermal fibroblasts (A1-F) and triple-negative breast cancer cells, (**B**) MDA-MB-468, and (**C**) MDA-MB-231 were treated with increasing concentrations of the fibronectin DAMP domains FnEDA or FnIII-1c alone or in combination for 4 h under serum-free conditions. Conditioned medium was collected after 4 h and IL-8 concentration determined by ELISA. IL-8 concentration shown represents the mean ± SEM of three independent experiments performed in triplicate.

**Figure 2 cells-11-02074-f002:**
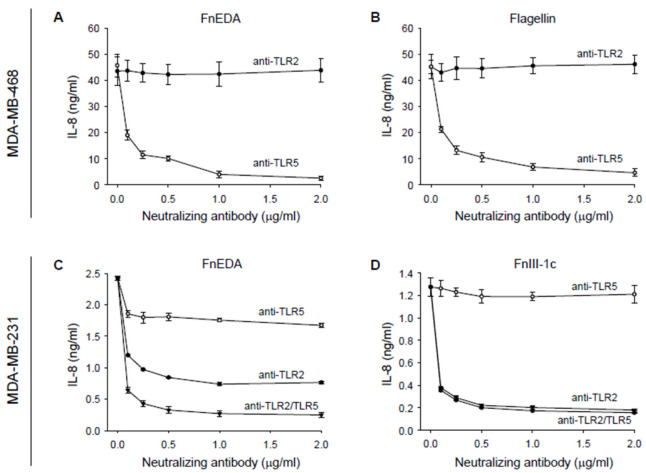
IL-8 synthesis stimulated by FnEDA and FnIII-1c in triple-negative breast cancer cells is TLR-dependent. (**A**,**B**) MDA-MB-468 and (**C**,**D**) MDA-MB-231 breast cancer cells were incubated with increasing concentrations of a neutralizing antibody to either TLR2 (anti-TLR2) or TLR5 (anti-TLR5) alone or in combination (anti-TLR2/TLR5) for 1 h prior to treatment with (**A**,**C**) FnEDA (5 µM), (**B**) the TLR5 agonist, flagellin (50 ng/mL), or (**D**) FnIII-1c (10 µM). After 4 h, conditioned medium was collected and IL-8 concentration determined by ELISA. Data shown represent the mean ± SEM of three experiments performed in triplicate.

**Figure 3 cells-11-02074-f003:**
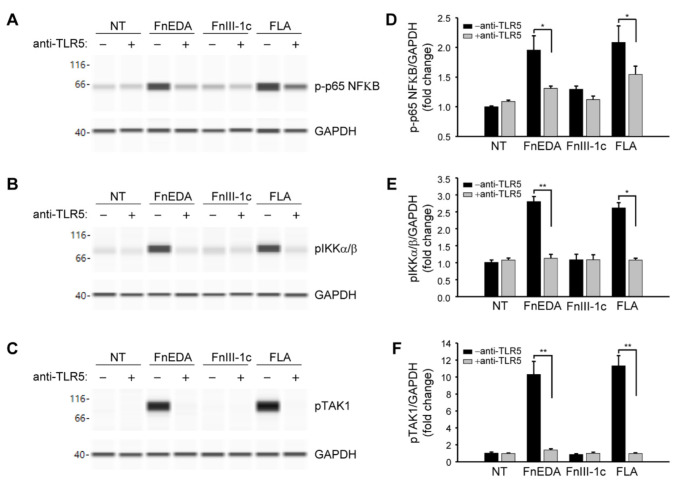
Activation of the NFκB signaling pathway by FnEDA in MDA-MB-468 breast cancer cells is TLR5-dependent. MDA-MB-468 breast cancer cells were incubated with a TLR5 neutralizing antibody (anti-TLR5; 2 µg/mL) for 1 h prior to treatment with either FnEDA (5 µM), FnIII-1c (10 µM), or flagellin (FLA, 50 ng/mL) for 1 h. Cell layers were then rinsed and whole-cell lysates analyzed by WES. Representative Wes-ProteinSimple assay for (**A**) phosphorylated p65 NFκB (p-p65 NFκB), (**B**) phosphorylated IKKα/β (pIKKα/β), and (**C**) phosphorylated TAK1 (pTAK1) are shown. Western analyses shown in (**A**–**C**) were quantified and normalized to GAPDH expression (**D**–**F**, respectively). Data represent the mean ± SEM of three independent determinations. NT indicates no treatment. * *p* < 0.05; ** *p* < 0.01; Student’s *t*-test.

**Figure 4 cells-11-02074-f004:**
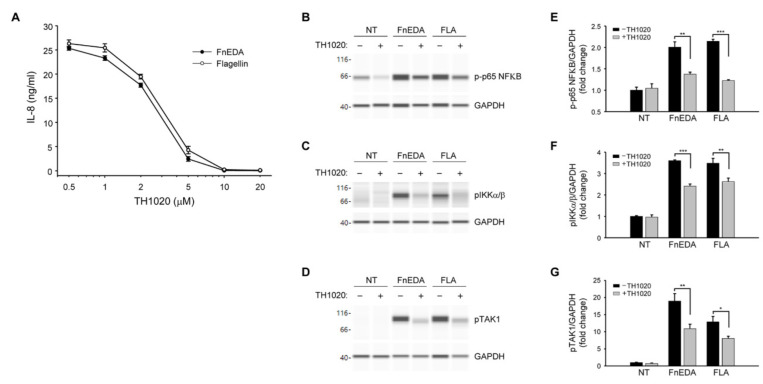
The TLR5 inhibitor, TH1020, blocks FnEDA-dependent IL-8 synthesis and NFκB signaling in MDA-MB-468 breast cancer cells. (**A**) MDA-MB-468 breast cancer cells were incubated with increasing concentrations of the TLR5 inhibitor, TH1020, for 1 h prior to treatment with either FnEDA (5 µM) or flagellin (50 ng/mL). After 4 h, IL-8 concentration in a conditioned medium was determined by ELISA. Data represent the average of three independent experiments performed in triplicate. To evaluate the effect of TH1020 on NFκB signaling, MDA-MB-468 breast cancer cells were incubated with TH1020 (10 µM) for 1 h prior to treatment with either FnEDA (5 µM) or flagellin (FLA, 50 ng/mL) for 1 h. Cell layers were rinsed and whole-cell lysates collected were analyzed by WES. Representative Wes-ProteinSimple assay for (**B**) phospho-p65 NFκB, (**C**) phospho-IKKα/β, and (**D**) phospho-TAK1 are shown. Western analyses shown in (**B**–**D**) were quantified and normalized to GAPDH expression (**E**–**G**, respectively). Data represent the mean ± SEM of three independent determinations. NT indicates no treatment. * *p* < 0.05; ** *p* < 0.01; *** *p* < 0.001; Student’s *t*-test.

**Figure 5 cells-11-02074-f005:**
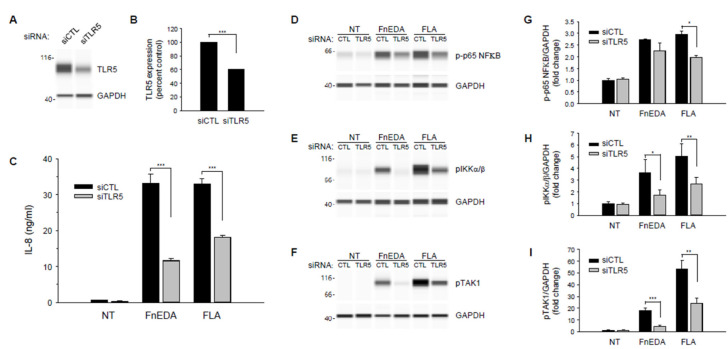
TLR5 knockdown suppresses FnEDA- and flagellin-induced IL-8 synthesis and NFκB signaling in MDA-MB-468 breast cancer cells. RNA interference was used to suppress expression of TLR5 (siTLR5) in MDA-MB-468 breast cancer cells. Non-targeting siRNA (siCTL) served as control. Representative Wes-ProteinSimple assay illustrating relative TLR5 expression following gene knockdown is shown in (**A**) and quantified in (**B**) relative to GAPDH expression (*n* = 3). (**C**) siRNA-treated MDA-MB-468 cells were incubated with either FnEDA (5 µM) or flagellin (FLA, 50 ng/mL) for 4 h and IL-8 concentration in conditioned medium determined by ELISA. Data shown represent the mean ± SEM of three experiments performed in triplicate. To evaluate the effects of reduced TLR5 expression on NFκB signaling, TLR5 knockdown cells were treated with either FnEDA (5 µM) or flagellin (FLA, 50 ng/mL) for 1 h. Whole-cell lysates collected were analyzed by WES. Representative Wes-ProteinSimple assay for (**D**) phospho-p65 NFκB, (**E**) phospho-IKKα/β, and (**F**) phospho-TAK1 are shown. Western analyses shown in (**D**–**F**) were quantified and normalized to GAPDH expression (**G**–**I**, respectively). NT indicates no treatment. * *p* < 0.05; ** *p* < 0.01; *** *p* < 0.001; Student’s *t*-test.

**Figure 6 cells-11-02074-f006:**
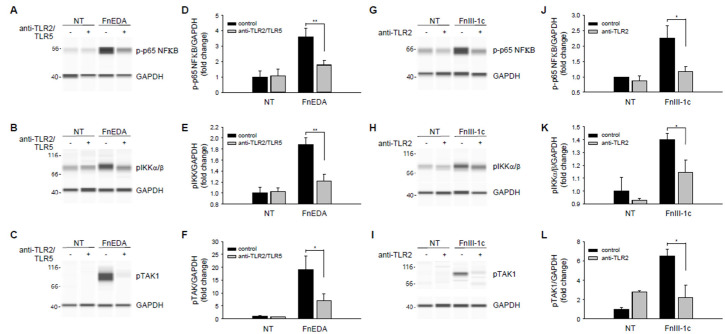
Effect of neutralizing antibodies to TLR2 and TLR5 on FnEDA- and FnIII-1c-induced NFκB signaling in MDA-MB-231 breast cancer cells. MDA-MB-231 breast cancer cells were incubated for 1 h with a TLR2 neutralizing antibody (2 µg/mL; anti-TLR2) alone or in combination with a TLR5 neutralizing antibody (2 µg/mL; anti-TLR2/TLR5) prior to treatment with either (**A**–**F**) FnEDA (5 µM) or (**G**–**L**) FnIII-1c (10 µM) for 1 h. Whole-cell lysates were collected and analyzed by WES. Representative Wes-ProteinSimple assay for (**A**,**G**) phospho-p65 NFκB, (**B**,**H**) phospho-IKKα/β, and (**C**, **I**) phospho-TAK1 are shown. Western analyses shown in (**A**–**C**,**G**–**I**) were quantified and normalized to GAPDH expression (**D**–**L**, respectively). Data represent the mean ± SEM of three independent determinations. NT indicates no treatment. * *p* < 0.05; ** *p* < 0.01; Student’s *t*-test.

**Figure 7 cells-11-02074-f007:**
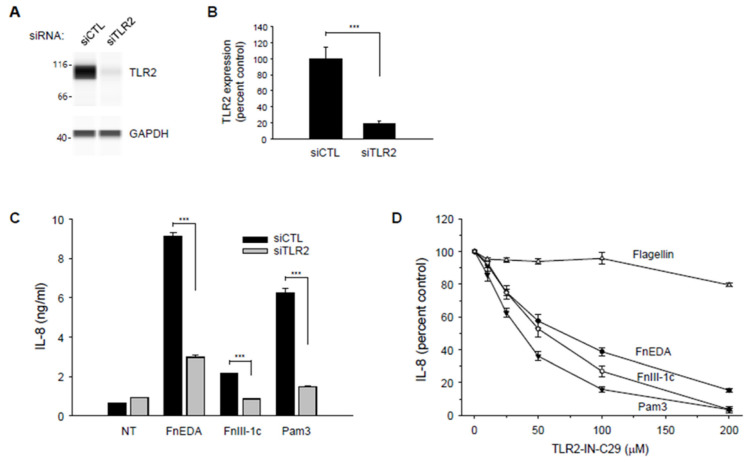
TLR2 knockdown suppresses FnEDA- and FnIII-1c-induced IL-8 synthesis in MDA-MB-231 breast cancer cells. RNA interference was used to suppress expression of TLR2 (siTLR2) in MDA-MB-231 breast cancer cells. Non-targeting siRNA (siCTL) served as control. Representative Wes-ProteinSimple assay illustrating TLR2 expression levels following TLR2 knockdown are shown in (**A**) and quantified and normalized to GAPDH expression (**B**, *n* = 3). (**C**) siRNA-treated MDA-MB-231 cells were incubated with either FnEDA (5 µM), FnIII-1c (10 µM), or the TLR2 agonist, Pam3CSK4 (Pam3, 0.1 μg/mL) for 4 h. Conditioned medium was collected, and IL-8 concentration determined by ELISA. Data represent the mean ± SEM of three independent experiments. (**D**) MDA-MB-231 cells were incubated with increasing concentrations of a TLR2 inhibitor (TLR2-IN-C29) for 1 h prior to treatment with either FnEDA (5 µM), FnIII-1c (10 µM), flagellin (50 ng/mL; TLR5 agonist), or Pam3 (0.1 ng/mL; TLR2 agonist) for 4 h. Conditioned medium was collected, and IL-8 concentration determined by ELISA. Data represent the average of three independent experiments carried out in triplicate. NT indicates no treatment. *** *p* < 0.001; Student’s *t*-test.

**Figure 8 cells-11-02074-f008:**
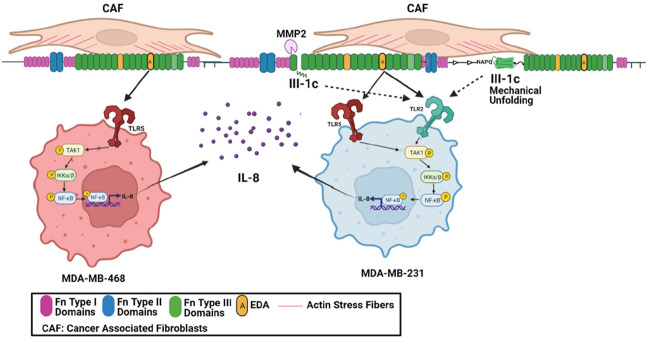
The impact of the tumor microenvironment on TLR activation in TBNC cells. Cancer-associated fibroblasts (CAFs) are highly contractile myofibroblasts which synthesize the EDA isoform of fibronectin and through the generation of mechanical force unfold the III-1 domain allowing the interaction of FnIII-1c with TLR2. In addition, the TNBC cells associate with the Fn matrix using integrin receptors and secrete MMP2 [63,64] which cleaves the fibronectin III-1 domain and frees FnIII-1c to interact with TLR2 on MDA-MB-231 cells. The EDA domain is also available to bind to TLR5 on MDA-MB-468 cells and to both TLR2 and TLR5 on MDA-MB-231 cells. Activation of the TLRs then leads to downstream signaling to activate TAK1 and IKKα/β and subsequent activation of NFκB. NFκB translocates to the nucleus where it acts as a transcription factor to regulate expression of IL-8. This graph was created using BioRender.

## Data Availability

Not applicable.

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
