# Peer review of "Fibronectin Functions as a Selective Agonist for Distinct Toll-like Receptors in Triple-Negative Breast Cancer"

_cells, 2022, doi:10.3390/cells11132074_

Round 1

Reviewer 1 Report

The manuscript by Ambesi and colleagues presents new findings regarding the effects of two DAMP (Damage Associated Molecular Pattern) domains of fibronectin on the release of the protumorigenic cytokine, IL-8, by two TNBC cell lines (MDA-MB-468 and MDA-MB-231).  Using well-characterized reagents and assays to directly test their hypotheses, novel data were generated demonstrating TLR5 (Toll-Like Receptor 5) activation by an ECM-derived DAMP in MDA-MB-468 cells, while fibronectin EDA (Extra Domain A) stimulation of IL-8 release in MDA-MB-231 cells depended on both TLR5 and TLR2.  Collectively, these new findings suggest TLR5 and TLR2 as possible therapeutic targets for the treatment of distinct subtypes of TNBC.  This study sheds new light on the molecular mechanisms underlying the role of the fibronectin matrix in breast cancer progression.

This is an exciting, complex, and well-conducted investigation.  The following comments are contributed to the authors for their consideration:

-          Please consider refining the title of the paper to a more succinct version.

-          Line 61, please change MDA-MD-468 to MDA-MB-468 and check other similar references.

-          It would be most helpful to create a schematic figure illustrating within the ECM associated with cells:

o   FnEDA, FnIII-1c

o   DAMPs

o   TLRs

o   IL-8

Author Response

  1. Please consider refining the title of the paper to a more succinct version.

      We have changed the title to “Fibronectin Functions as a Selective Agonist for Distinct Toll-like    Receptors in Triple Negative Breast Cancer Cells”.

  1. Line 61, please change MDA-MD-468 to MDA-MB-468 and check other similar references.

      This has been corrected.

  1. It would be most helpful to create a schematic figure illustrating within the ECM associated

      cells:  FnEDA, FnIII1-c, DAMPs, TLRs, IL-8.

      We have added such a figure to the manuscript as Figure 8.

Reviewer 2 Report

The manuscript “Fibronectin-Derived Damage Associated Molecular Pattern Molecules Promote an Inflammatory Microenvironment in Triple Negative Breast Cancer by Serving as Selective Agonists for Toll-Like Receptors” follows previous publications of Dr. McKeown-Longo’s lab. Even thought it highlights a novel PRR involved in Fibronectin EDA recognition and signalling, TLR5, is not clear why this is. Several previous studies by the authors and others have shown the importance of TLR4, and also TLR2, in the recognition of Fn-EDA, in the context of different diseases including cancer. Here using two breast cancer cell lines, the authors show the importance of TLR5 and point towards a cell specific response. However, many different questions arise from the data shown and some key points should be experimentally addressed.

Moreover the “Data not shown” (line 171), should be included in the manuscript, showing the response of both cell lines after treatment with neutralizing antibodies for different TLRs. Is also important to show that the recognition and IL-8 activation occurs in the presence of regular agonist, as shown for TLR5 with flagellin. This is important to exclude the possibility that in certain cell lines this activation does not occur for specific TLRs because they may not be fully functional.

In order to understand the cell specificity and if there is any specific link between the authors findings and the cell lines used being TNBC, the authors should repeat data on Figures 1 and 2 for additional cell lines, including some other breast cancer cell lines and other cancer cell types. The title of the manuscript should be adjusted or tunned down according to these new experiments.

Specific comments

The sentence on the lines 20-22 (abstract) lacks a conclusion. Why do the authors highlight this difference between cell lines in the abstract? What to conclude about it?

Line 44 – the sentence lacks reference (ex PMID: 11150311)

Line 44-46 – definition of DAMPs should be improved. DAMPs are not only recognized by TLRs, but by a variety of PRRs.

Line 60 – The reason to look for the impact on IL-8 should be clarified.

Author Response

  1. “Even though it highlights a novel PRR involved in Fibronectin EDA recognition and signalling, TLR5, it is not clear why this is”.   

We have now addressed this issue in the Discussion.

  1. “It is important to show that the recognition and IL-8 activation occurs in the presence of regular agonist”.  

      The MD 468 cells do not respond to LPS.   We do not know the molecular basis for this, but we now address this in the Discussion.

  1. “The authors should repeat the data….for additional cell lines…and other cancer cell types”.

We agree that this is an interesting and important question and will be addressing it in future studies. 

Specific Comments:

  1. The sentence on the lines 20-22 (abstract lacks a conclusion). Why do the authors highlight this difference between cell lines in the abstract?  What to conclude about it? 

      We have rewritten this section of the abstract.

  1. Line 44- the sentence lacks reference.

      We have added two new references.

  1. Line 44-46-Definition of DAMPs should be improved. DAMPs are not only recognized by TLRs, but by a variety of PRRs.

      We have included this information and two new references for PRRs.

  1. Line 60- The reason to look for the impact on IL8 should be clarified.

      We have expanded this section of the Introduction and cited additional recent references.   

Round 2

Reviewer 2 Report

The authors should either include additional TNBC cell lines or tune down their conclusions. In this last option, the focus should be on the common points between the two TNBC cell lines presented in the study.

Author Response

The authors should either include additional TNBC cell lines or tune down their conclusions.  In this last option, the focus should be on the common points between the two TNBC cell lines presented in the study.

It seems to me that the reviewer’s main concern is that we should further support our conclusion by expanding our studies to include additional TNBC cell lines.  While appreciating the value of this suggestion, these additional experiments would take several months to complete. 

I assume that he/she is reacting to our statement at the end of the abstract “the data point to TLR5 and TLR2 as possible therapeutic targets to treat distinct subtypes of TNBC.”  We also have a similar statement at the end of the introduction “….the data suggest that TLRs may provide an opportunity for the development of subtype specific targets for the treatment of TNBC.” 

To address this issue, I have removed these two sentences from the manuscript and left the speculation about potential therapeutic targets for the Discussion.